# Antimicrobial and Digestive Effects of *Yucca schidigera* Extracts Related to Production and Environment Implications of Ruminant and Non-Ruminant Animals: A Review

Aracely Zúñiga-Serrano [1], Hugo B. Barrios-García [1], Robin C. Anderson [2], Michael E. Hume [2], Miguel Ruiz-Albarrán [1], Yuridia Bautista-Martínez [1], Nadia A. Sánchez-Guerra [1], José Vázquez-Villanueva [1], Fidel Infante-Rodríguez [1] and Jaime Salinas-Chavira [1,*]

1   College of Veterinary Medicine and Animal Science, Universidad Autónoma de Tamaulipas, Ciudad Victoria 87000, Mexico
2   Food and Feed Safety Research Unit, Southern Plains Agricultural Research Center, Agricultural Research Service, United States Department of Agriculture, College Station, TX 77845, USA
*   Correspondence: jsalinasc@hotmail.com

**Abstract:** Plant extracts have been used over time in traditional medicine, mainly for their antimicrobial activity as well as for their medicinal effects. Plant-derived products contain secondary metabolites that prevent pathogenic microbial growth similar to conventional medicines. These secondary metabolites can enhance animal health and production in a more natural or organic manner and may contribute to the reduction in the use of pharmacological drugs in animal feed, which is of great concern for emerging microbial resistance. Plant secondary metabolites can be cost effective, while improving the production efficiency of ruminants, non-ruminants, and aquatic food animals. Among the plant-derived products is the *Yucca schidigera* extract (YSE), containing steroidal saponins as their main active component. YSE has multiple biological effects, including inhibition of some pathogenic bacteria, protozoa, and nematodes. YSE is used to control odor and ammonia and consistently enhance poultry production by enhancing intestinal health and function. In pigs, results are as yet inconclusive. In ruminants, YSE works against protozoa, has selective action against bacteria, and reduces the archaea populations; all these effects are reflected in the reduction in emissions of polluting gases, mainly methane, although the effects are not observed in all feeding conditions. These effects of YSE are discussed in this review. YSE has potential as a natural feed additive for sustainable animal production while contributing to the mitigation of contaminant gas emissions.

**Keywords:** *Yucca schidigera*; antimicrobial; secondary metabolites; sustainability; pollution; production; food animals

## 1. Introduction

For several years, antibacterial drugs have been routinely used in diets to enhance food animal production efficiency in intensive systems; nevertheless, there is concern about increased bacterial resistance to these drugs [1,2]. As a consequence, alternative products are sought to prevent the proliferation of pathogenic microorganisms, as well as those that can enhance food animal production. Among these alternatives are plant extracts that may offer fewer adverse effects than conventional drugs and have similar or superior effects against pathogenic microorganisms to those of traditional drugs [3–5].

Much of the research on natural products to prevent or control infections is primarily geared towards plant extracts and essential oils, several of them being related to phenolic compounds, terpenes, saponins, tannins, flavonoids, organic acids, complex carbohydrates, and non-protein amino acids [6–8]. Antimicrobial effects of plant secondary metabolites

include cell membrane disruption, enzyme inhibition, substrate deprivation, and prevention of bacterial colonization. Plant extracts can contain many different secondary metabolites that can have a synergistic biological effect when used in combination with other plant extracts or with other feed additives used to promote growth in livestock, such as probiotics and organic acids [9,10]. The antimicrobial compounds obtained from secondary metabolites of plants have potential use in ruminants, non-ruminants, and aquatic organisms [11] and are a real alternative to traditional antibiotics to improve the health, productive performance, and quality of food animals.

Different research works are being conducted for agricultural systems sustainability [12,13]. In this aspect, diet formulations must provide nutrients for optimal growth of animals, and the inclusion of additives extracted from plants has potential as a common practice to enhance animal production efficiency [14]. Among these extracts is the one obtained from *Yucca schidigera* (YSE), which has antibacterial, antiprotozoal, and antifungal effects [15–17]. YSE contains saponins and polyphenols; however, the main biological effects are associated with the steroidal fraction of saponins [18–23]. The saponins trap ammonia-nitrogen, reducing odor and ammonia emissions in pig and poultry production [10,24,25], as well as demonstrating influence on cat and dog fecal metabolites and microorganisms [26] and in reducing ammonia accumulations in aquaculture environments [27–30].

In ruminants, feed supplementation with YSE reduces gas emissions that pollute the environment. The positive effects of YSE on methane and ammonia mitigation in livestock are well documented [31]. The saponins in YSE reduce the methanogenic archaea (methanogens) in the rumen, mainly reducing the protozoa associated with methanogens [32]. Additionally, YSE also inhibits the growth of some rumen bacteria and fungi [15].

There is information available regarding the effects of YSE on the different enteric microorganisms in ruminants and in non-ruminants [25,33–35]; also, there are some scientific reports about the use of YSE to enhance animal production efficiency and to reduce ambient pollution [36,37]. All of this information must be integrated and organized into an approach to maximize the potential of YSE for sustainable animal production, reduce conventional drug use as feed additives, and reduce polluting gas emissions. It is also important to show those areas that need more research because some aspects of efficacy and application are not conclusive. The objective of the present study is to review the antimicrobial and digestive effects of *Yucca schidigera* extracts related to the production and environmental implications of ruminant and non-ruminant food animals.

## 2. *Yucca schidigera*, Distribution, and General Characteristics

*Yucca schidigera* is a native plant in the states of Baja California and Sonora, Mexico; its habitat extends from the arid region of northwestern Mexico to the Mojave Desert in the southwestern United States. *Yucca schidigera*, with the common names Mojave yucca, Mohave yucca, and Spanish dagger, is a shrub or small tree that belongs to the Agavaceae family. It is a slow-growing plant that thrives in a climate characteristic of high temperature and low rainy precipitation [38]. *Yucca schidigera* has medicinal properties and was used by Native Americans to treat various diseases, one of them being arthritis [39]. The plant has straight leaves and white, bell-shaped, edible flowers and bears edible fruits [40]. The *Yucca schidigera* products are approved for human use by the FDA [39]. To obtain extracts, the plants are harvested and transported to the factory, where the trunks are mechanically processed to obtain the extracts [41]. Currently, the plant is used in animal feed with multiple benefits enhancing animal production efficiency and acting against multiple harmful organisms such as protozoa, nematodes, and bacteria (mainly Gram-positive). *Yucca schidigera* contributes to reducing ammonia and methane emissions in confined animals. In addition, it has other uses for commercial purposes such as in beverages and agriculture [39]. Yucca is used to control nematodes and fungi in agricultural crops [42].

### 2.1. Saponin in Yucca schidigera Extracts

Saponin is the main active compound in YSE. The word saponin comes from Latin "sapo" that in English is "soap" because it forms foams like soaps with water; saponins have surfactant properties due to their amphiphile nature. In the chemical structure of saponins are observed two components: one is the hydrophobic aglycone (sapogenin) bounded to the second part that is the hydrophilic component composed of carbohydrates such as glucose, fructose, galactose, and arabinose [43]. According to Francis et al. [43] and Tamura et al. [44], the aglycone component is used for the classification of saponins into two groups: steroidal (steroid saponins) or triterpenoid (triterpene saponins). The steroidal saponins may contain smilagenin and sarsasapogenin. Cheeke [41] recognized that the main plants used for industrial extraction of saponins are *Yucca schidigera* (from Mexico) and *Quillaja Saponaria* (from Chile), which have steroidal (Yucca) and triterpenoid (Quillaja) structures, respectively; however, saponins are also found in many other plants [43].

Both steroid and triterpene saponins have been explored for many years with commercial applications. Due to their detergent properties, saponins have exhibited several biological activities of economic importance with applications in the pharmaceutical industry, foods, cosmetics, and beverages [43,44]. In animal production, the commercial applications of saponins in YSE are in the cereal grain processing known as tempering [45,46] and for odor and pollution control from poultry and pigs' feces. Matusiak et al. [25] observed in poultry manure that YSE had effective control of odorous and volatile compounds such as ammonia, hydrogen sulfide, dimethylamine, trimethylamine, and isobutyric acid; they observed greater effect when combining YSE with other microbial treatments.

Extracts from the *Yucca schidigera* plant show antimicrobial effects against yeasts and dermatophytes, as well as antibacterial and antiprotozoal properties, and have been used for many years in food production and cosmetics [47]. However, it has been reported that YSE has no effect on the concentration of microorganisms under specific conditions [48], and the growth of microorganisms depends on the substrate and the concentration of microbiota present. Thus, the implications for effects on the gut microbiota differ according to their ecological niche [33].

The presence of saponins in plants appears to be linked to defensive functions against pathogenic bacteria and fungi [9,11,49]. In animals, saponins inhibit microbial growth by adsorption to the microbial cell membranes, disruption of cytoplasmic membranes, and cell leakage [33,50]. However, these inhibitory effects of saponins are seen at lower population densities and may not be effective with higher densities [33]. Some bacteria, including *Fusobacterium necrophorum* and *Clostridium perfringens*, have reduced growth when exposed to culture medium containing 10% YSE. In addition, YSE has reduced rumen protozoa numbers in cattle in in vitro and in vivo studies [15,17,22,51,52]. Extracts of *Yucca schidigera* eliminated *Giardia* trophozoites in in vitro studies; however, these extracts, when included in diets of gerbils and lambs, did not alter the course of experimentally induced giardiasis but did diminish the excretion of cysts [17].

### 2.2. Influence of Yucca schidigera Extracts on Enteric Microorganisms

The influence of YSE on microbial growth in different conditions of study is shown in summary in Table 1. According to this table, YSE has different effects on microbial growth. Katsunuma et al. [34], in in vitro studies with 20 strains of bacteria isolated from the intestinal tract of animals, observed that *Yucca schidigera* extract did not inhibit the growth of seven strains of bacteria (*Lactobacillus plantarum, Lactobacillus rhamnosus, Enterococcus hirae, Escherichia coli, Bifidobacterium thermophilum, Bifidobacterium longum*, and *Streptococcus bovis*); however, the other 13 strains (*Bacteroides fragilis, Fusobacterium varium, Fusobacterium necrophorum, Clostridium perfringens, Clostridium innocuum, Clostridium sporogenes, Veillonella parvula, Propionibacterium acnes, Eubacterium aerofaciens, Selenomonas ruminantium, Peptococcus asaccharolyticus, Ruminococcus productus*, and *Megasphaera elsdenii*) were inhibited by YSE. It is interesting to note that the antimicrobial effects of saponins from *Yucca schidigera* are due to the compounds sarsasapogenin and smilagenin, present in the butanol-extractable

fraction. Conversely, the aqueous non-butanol-extractable fraction of *Y. schidigera* saponins influences various effects on in vivo nitrogen metabolism [33,35].

Matusiak et al. [25] observed that 5% YSE reduced most populations of several pathogenic microorganisms, including *E. coli., Listeria monocytogenes, Salmonella Typhimurium,* and *Enterococcus faecalis.* In addition, 15% YSE reduced all potentially pathogenic microorganisms. The potentially beneficial lactic acid bacteria *Leuconostoc mesenteroides* and *Lactiplantibacillus plantarum* were not influenced by YSE.

### 2.3. Influence of Yucca schidigera on the Growth Microorganisms of Poultry

*Yucca schidigera* extract has the potential to decrease the number of pathogenic microorganisms or those that may reduce the productive performance of poultry. In addition, YSE may not influence the growth of beneficial bacteria such as those producing lactic acid. More research is warranted in this aspect, mainly using YSE plus probiotics and prebiotics. Ayoub et al. [53] supplemented YSE to broiler chickens in drinking water and observed reductions of intestinal bacteria for total bacteria counts and *E. coli*; however, lactic-acid-producing bacteria numbers were not influenced. Alghirani et al. [54] tested the oxytetracycline antibiotic or *Yucca schidigera* at different levels in broiler feed. In all treatments, they observed in cecal contents the presence of *Escherichia coli* and *Bacillus* sp.; however, *Enterococcus faecalis* growth was inhibited by the antibiotics and by *Yucca schidigera*. Bafundo et al. [55] observed reductions in mortality of broiler chickens, reduced numbers of fecal Clostridia, and decreased percentages of broiler chickens with *Salmonella* when treated with a product obtained from *Quillaja saponaria* trees and *Yucca schidigera* plants. In another study, *Yucca schidigera* additions to the litter of broiler chickens did not influence total colony counts of Enterobacteriaceae, yeast, mold, pH, moisture, or ammonia-N [56]. A blend of caprylic acid plus *Yucca schidigera* extract consistently reduced *E. coli* and had no effect on *Lactobacillus* in laying hens [57] or broiler chickens [58] but had no effect on the growth of *Clostridium perfringens* and *Bifidobacteria*.

### 2.4. Influence of Yucca schidigera on the Growth Microorganisms of Pigs

Weaning is critical due to the stress of piglets. In this stage, a decrease in feed intake, an increase in morbidity, and enteric infections with diarrhea are observed [59]. Diarrhea is of economic importance in pig production [60]. As mentioned, YSE may reduce or inhibit the growth of enteric microorganisms with pathogenic potential. In this sense, Yang et al. [61] supplemented weaning piglet diets with YSE and observed reduced diarrhea rate and mortality and increased diversity and abundance of cecal microflora. At the family level, the authors saw reduced numbers of Bacteroidales, Clostridiaceae, Veillonellaceae, Erysipelotrichaceae, Acidaminococcaceae, Streptococcaceae, Campylobacteraceae, and Streptomycetaceae. The authors also mentioned that YSE reduced the growth of bacteria associated with ammonia production in the intestine. Katsunuma et al. [62] observed with pig YSE-supplemented diets that there was no change in total fecal counts of viable microorganisms. However, Bifidobacteria, eubacteria, and staphylococci were more abundant, while the *Veillonella* number was lower in the feces of pigs supplemented with YSE. More research is warranted on effect of YSE on other pathogenic microorganisms such as those of the Enterobacteriaceae family, as well as on normal or beneficial microorganisms found in the gastrointestinal tract of healthy pigs.

### 2.5. Influence of Yucca schidigera on Microbial and Fermentation in Ruminants

The possible use of YSE to decrease gas emissions that pollute the environment and that are produced by livestock was reviewed by Adegbeye et al. [63]. YSE mainly reduced methane and nitrous oxide, as well as urinary and fecal nitrogen excretion. The authors documented that saponins as the main active compounds in YSE reduced ruminal cellulolytic bacteria and fungi that produce the contaminant gases. In addition, Jayanegara et al. [64], in a meta-analysis study, established that the increase in saponin levels reduced methane and acetate levels and protozoal counts; however, propionate was increased with

increased saponin levels. They also observed greater reductions in methane levels for yucca than for tea or quillaja as sources of saponins. Sun et al. [31] recognized the beneficial effects of YSE on methane and ammonia mitigation in in vitro studies; however, in some studies, YSE did not influence methane or ammonia production, attributing those differences to different experimental conditions.

The influence of YSE on microbial and ruminal fermentation characteristics is shown in summary in Table 2. In this table, the different factors that influence the effect of YSE on rumen fermentation can be observed. Hristov et al. [36], in beef heifers supplemented with YSE, did not observe an influence on ruminal pH or degradability of peptides; amino acids; microbial protein synthesis; or digestibility of crude protein (CP), dry matter (DM), or neutral detergent fiber (NDF). However, they did observe reduced ruminal ammonia, protozoa numbers, and an increase in propionate concentration with YSE supplementation. Consistently, Liu and Li [37], in ruminal cannulated lambs fed a basal diet supplemented with YSE, did not observe changes in ruminal pH, but ruminal ammonia concentrations and protozoa populations were decreased, and they saw increased ruminal propionate concentrations. In addition, the authors observed reduced acetate levels and improved digestibility of DM, CP, and NDF with YSE supplementation.

Eryavuz and Dehority [48] found that in lambs, YSE supplementation did not influence bacteria (total and cellulolytic) and fungal concentrations. Lambs given YSE at 30 g/head/day had higher protozoal numbers and pH in rumen contents. With different results, Wang et al. [15], in in vitro studies with ruminal microorganisms, observed that saponin extracted from *Yucca schidigera* had different effects on the growth of amylolytic bacteria, decreasing the numbers of *Prevotella bryantii*, *Ruminobacter amylophilus*, and *Streptococcus bovis* but increased *Selenomonas ruminantium*. The authors also observed that saponins from *Yucca schidigera* inhibited the growth of cellulolytic bacteria and fungi.

Narvaez et al. [65], using in vitro fermentations of a barley-based diet supplemented with monensin alone or combined with extracts of hops and YSE, found that all treatments reduced CH4, NH3-N, microbial protein, and propionate. In addition, YSE increased total VFA production. Monensin reduced *Ruminococcus flavefaciens* and increased *Selenomonas ruminantium*, while both microorganisms increased with only YSE supplementation, as did *Fibrobacter succinogenes* and *Ruminococcus albus*. The methanogenic archaea were decreased by all treatments. YSE has multiple effects in rumen related to methane reductions (Figure 1). The influence of YSE on other ruminal changes, such as on metabolites and on different ruminal microorganisms, is not complete, and research on these themes is warranted.

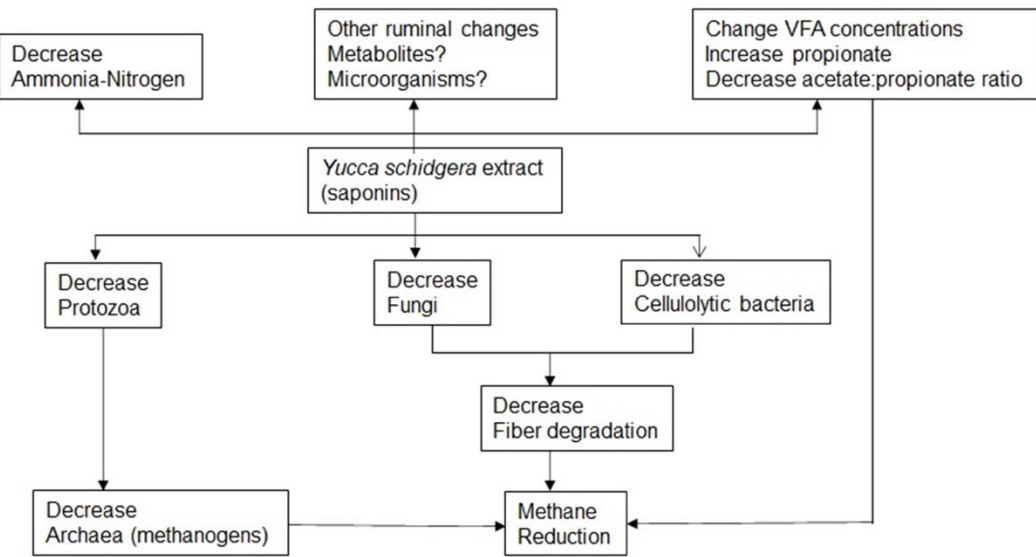

**Figure 1.** Proposed modes of action for ruminal methane reduction with *Yucca schidigera* extracts.

The beneficial effects of YSE on rumen fermentation are not always observed. Canul-Solis et al. [66], in lambs fed *Pennisetum purpureum* grass, did not find effects of saponins from *Yucca schidigera* on DM intake and digestibility of DM, OM, NDF, or production of VFA or CH4. Similarly, Galindo et al. [67], using in vitro ruminal incubations with star grass *(Cynodus nlemfuensis)* as substrate and saponin extract from *Sapindus saponaria*, observed reductions in protozoa numbers but increased methanogen numbers and methane production. Wang et al. [68] observed that YSE improved in vitro fermentation of barley grain but not with alfalfa hay. In addition, the fermentation of barley grain resulted in increased propionic acid and VFA; however, ammonia production was decreased. These results may indicate that the beneficial influence of YSE could be expected with high grain-based diets. Furthermore, Guyader et al. [69] reported that saponins caused reductions in protozoa populations and methane production in in vitro studies with ground cereal grains as substrate. However, in in vivo studies with dairy cows fed a diet with 60% forage, the same saponin did not mitigate methane production. Because several factors account for methane and ruminal fermentation characteristics, more research is warranted to elucidate the role of saponins or YSE on methane production and feed efficiency in in vivo studies under different feeding conditions.

**Table 1.** Effect of *Yucca schidigera* extract (YSE) on microbial growth in different conditions of research.

| Microorganism | Conditions of the Study | Reference |
|---|---|---|
| **Reports Where YSE Inhibited Microbial Growth** | | |
| *Enterococcus faecalis* | Inhibited by antibiotics and by YSE. Study in broiler chickens | Alghirani et al. [54] |
| *Bacteroides fragilis; Fusobacterium varium; Fusobacterium necrophorum; Clostridium perfringens; Clostridium innocuum; Clostridium sporogenes; Veillonella párvula; Propionibacterium acnés; Eubacterium aerofaciens; Selenomonas ruminantium; Peptococcus asaccharolyticus; Ruminococcus productus; Megasphaera elsdenii* | In vitro study | Katsunuma et al. [34] |
| *E. coli; Listeria monocytogenes; Salmonella Typhimurium; Enterococcus Faecalis* | 15% YSE reduced all potentially pathogenic microorganisms | Matusiak et al. [25] |
| *Candida famata; Pichia carsonii; Pichia nakazawae; Dermatomyces hansenii; Zygosaccharomyces* | In vitro study | Miyakoshi et al. [16] |
| **Reports where YSE reduced microbial growth** | | |
| *E. coli* | A blend of caprylic acid plus YSE | Wang et al. [57] |
| *E. coli* | Cecal microbiota in chickens | Begum et al. [58] |
| *Neocallimastix frontalis; Piromyces rhizinflata* | In vitro study. Ruminal fungi | Wang et al. [68] |
| *Prevotella bryantii; Ruminobacter amylophilus; Streptococcus bovis* | In vitro study. Cellulolytic ruminal bacteria | Wang et al. [68] |
| *Bacteroidales; Clostridiaceae; Veillonellaceae; Erysipelotrichaceae; Acidaminococcaceae; Streptococcaceae; Campylobacteraceae; Streptomycetaceae* | Bacterial Family. Study with piglets | Yang et al. [61] |
| *Salmonella* | Reductions in mortality of broiler chickens | Bafundo et al. [55] |
| *Giardia trophozoite* | Trophozoite in the ileum | McAllister et al. [17] |
| *Escherichia coli* | Study in broiler chickens (intestinal bacteria) | Ayoub et al. [53] |
| *Clostridia* spp. | bacterial fecal | Bafundo et al. [55] |
| *Veillonella* | Study with pig feces | Katsunuma et al. [62] |

**Table 1.** *Cont.*

| Reports Where YSE Inhibited Microbial Growth | | |
|---|---|---|
| **Reports where YSE had no effect on microbial growth** | | |
| *Lactobacillus plantarum; Lactobacillus rhamnosus Enterococcus hirae; Escherichia coli; Bifidobacterium thermophilum; Bifidobacterium longum; Streptococcus bovis* | In vitro study | Katsunuma et al. [34] |
| *Leuconostoc mesenteroides; Lactiplantibacillus plantarum* | Lactic acid bacteria important in nutrition | Matusiak et al. [25] |
| *Lactobacillus* | Laying hens | Wang et al. [57] |
| *Lactobacillus; Bifidobacteria; Clostridium perfringens* | Broiler chickens | Begum et al. [58] |
| *Lactobasullus; Streptococcus; Staphylococcus* | Study with pig feces | Katsunuma et al. [62] |
| Enterobacteriaceae *E. coli* | Study in litter of broiler chickens In vitro study | Onbasilar et al. [56] Killeen et al. [33] |

**Table 2.** Effects of *Yucca schidigera* extracts (YSE) on microbial and ruminal fermentation characteristics.

| **Effect of *Yucca schidigera* Extracts** | **Study Characteristics** | **Reference** |
|---|---|---|
| YSE reduced ruminal ammonia (33.5%) and protozoa numbers (20.3%). Increased propionate concentration (18.2%). | Beef heifers fed a diet with 61% barley grain and 39% alfalfa silage (DM basis) | Hristov et al. [36] |
| YSE reduced ruminal ammonia (14.42%), protozoa (17.3%), and acetate (17.5%). Improved propionate (16.9%) and digestibility of DM, CP, and NDF. | Lambs fed a diet of 50:50 forage to concentrate | Liu and Li [37] |
| YSE increased protozoal numbers (23.3%) and pH (6.5%) in rumen. YSE did not influence total or cellulolytic bacteria and fungi. | Lambs fed a diet with 69.5% concentrate and 30.5% alfalfa meal (DM basis) | Eryavuz and Dehority [48] |
| YSE had different effects on amylolytic bacteria. Decreased numbers of *Prevotella bryantii, Ruminobacter amylophilus,* and *Streptococcus bovis* but increased *Selenomonas ruminantium.* YSE inhibited the growth of cellulolytic bacteria and fungi. | Pure culture in in vitro studies | Wang et al. [15] |
| YSE reduced methane (55.3%), ammonia (11.9%), microbial protein (39.9%), acetate (16.8%), total VFA, and methanogenic archaea (23.9%). YSE increased propionate (67%), as well as the microorganisms *Ruminococcus flavefaciens, Selenomonas ruminantium, Fibrobacter succinogenes,* and *Ruminococcus albus.* | In vitro study with barley grain as substrate | Narvaez et al. [65] |
| YSE had no effect on DM intake, digestibility of DM, OM, NDF, or on the production of VFA or methane | In lambs fed *Pennisetum purpureum* grass | Canul-Solis et al. [66] |
| YSE improved in vitro fermentation of barley grain but not with alfalfa hay. Fermentation of barley grain resulted in increased propionic acid and VFA; however, ammonia production was decreased. | In vitro fermentation | Wang et al. [68] |
| Saponins caused reductions in protozoa populations and methane production. However, in in vivo studies with dairy cows fed a diet with 60% forage, the same saponin did not mitigate methane production. | In in vitro studies with ground cereal grains as substrate. | Guyader et al. [69] |

*2.6. Effects of Yucca schidigera Supplementation on Production Behavior and Carcass Traits of Broiler Chickens*

The enhanced production parameters of broiler chickens supplemented with YSE are well documented (Table 3), revealing improved carcass characteristics and meat composition. Other beneficial effects of YSE supplementation were improved antioxidative and immune capability, as well as improved nutrient digestibility. Sahoo et al. [70] reported that YSE supplementation in broiler chickens improved average weight gain (9.7%), feed conversion ratio (9%), survivability (3.5%), eviscerated weight yield (8.1%), breast yield (6.2%),

and thigh yield (11.4%). All of these improvements were reflected in higher economic profit with YSE supplementation.

**Table 3.** Effects of *Yucca schidigera* supplementation on production behavior and carcass traits of broiler chickens.

| Effect of YSE on Broiler Productive Variables | Conditions of the Study | Reference |
|---|---|---|
| YSE improved average weight gain (9.7%), feed conversion ratio (9%), survivability (3.5%), eviscerated weight yield (8.1%), breast yield (6.2%), and thigh yield (11.4%). | Broiler chickens in winter season | Sahoo et al. [70] |
| YSE had no effect on weight gain. It improved feed conversion ratio (13.5%), protein efficiency (19.5%), and reduced feed intake (11.4%). | YSE in drinking water of broiler chickens in finisher phase | Ayoub et al. [53] |
| YSE enhanced feed efficiency in the whole experimental period. | Broiler chickens | Sun et al. [31] |
| The product of Quillaja and Yucca at 42 d enhanced FCR (5.4%) and reduced mortality (79%). | Broiler chickens with a product of *Quillaja saponaria* and *Yucca schidigera*. | Bafundo et al. [55] |
| At 42 d, YSE improved body weight gain (15.2%), FCR (9.4%), carcass weight (7.1%), and breast weight (2.05%). | Broiler chickens in a tropical environment | Alghirani et al. [54] |
| At 35 d, YSE plus caprylic acid improved body weight gain (10.7%) and FCR (5.6%). The effect was small on carcass characteristics. | *Yucca schidigera* extract combined with caprylic acid | Begum et al. [58] |

Ayoub et al. [53], supplementing YSE in the drinking water of broiler chickens, observed enhanced feed conversion, antioxidative biomarkers, and immunoglobulin levels; in addition, litter nitrogen and water concentration were reduced. Sun et al. [31], with YSE supplementation, saw in broiler chickens enhanced production performance during the finisher period at grow out, as well as improved liver antioxidative capability.

Bafundo et al. [55] examined disease response in broiler chickens given a product of *Quillaja saponaria* trees and *Yucca schidigera* plants. They observed reductions in mortality of birds and reductions in numbers of fecal Clostridia and *Salmonella* colonization. In broiler chickens produced in a tropical environment, Alghirani et al. [54] observed that YSE improved production parameters, digestibility of nutrients, gut health, carcass yields of main cuts, and meat chemical composition through increased CP, reduced crude fat, and saturated fatty acids. Begum et al. [58] showed that *Yucca schidigera* extract combined with caprylic acid improved the growth performance of broilers. The improved production variables were evident during the finisher phase and at grow out. The effect was small on carcass characteristics. In broiler chickens supplemented with probiotics plus YSE, Benamirouche et al. [71] observed improved meat chemical composition. They saw increased pH, crude protein, and reduced crude fat levels. In addition, saturated fatty acids were reduced, and polyunsaturated fatty acids were increased with the supplements.

There is limited information about the effects of YSE on the coccidia of broiler chickens. Rodríguez et al. [72] reported that broiler chickens challenged with coccidia and supplemented with YSE saponin and *Trigonella foenum-graecum* showed reduced oocyst counts and intestinal lesions, and these improvements were reflected in enhanced production performance of the birds. These results agree with Bafundo et al. [73] using saponins of quillaja plus yucca. Alfaro et al. [74] reported the synergic influence of YSE and coccidiosis vaccine on the productive performance of broiler chickens; however, YSE did not have the same positive effect when combined with the coccidiostat monensin.

*2.7. Effect of Yucca schidigera Supplementation on Production Behavior and Carcass Traits of Pigs*

Yang et al. [61] observed in weaned piglets supplemented YSE or *Candida utilis* that YSE improved production performance and reduced diarrhea. There was also enhanced immunity, antioxidant function, and intestinal health. The improved production performance with YSE and *Candida utilis* was quite similar, although no synergic effect was detected.

In agreement, Fan et al. [75], in weaned piglets, also reported that YSE improved feed efficiency and reduced hindgut NH3-N production with increased nutrient digestibility and reduced blood urea concentrations. Espinosa-Muñoz et al. [14], in growing and finishing pigs supplemented with YSE in diets, also found decreased blood urea, triglycerides, and cholesterol. In contrast, Gebhardt et al. [76], in growing finishing pigs, found a very marginal advantage of YSE combined with chromium propionate on the production performance. The treatments did not influence carcass characteristics. These differences may be related to the initial body weights of the animals, and other factors related to management; the available information is limited as to document these effects, and research is warranted on these themes.

## 3. Conclusions

In addition to odor and ammonia control, YSE can be used to enhance animal production efficiency and contributes to diminishing ambient pollution mainly in confined animals. YSE may decrease or inhibit the growth of some pathogenic microorganisms that reduce animal production efficiency. The most consistent effect of YSE is in poultry production enhancement. In pig production, more research is warranted to identify the stage of production (from weaning to finishing and for animals in reproduction) where YSE supplementation could be more effective to improve the health and production of animals. More information is required for poultry and pig production regarding the effect of YSE combined with other feed additives such as probiotics or prebiotics. In ruminants, the effects of YSE can be influenced by diet type; however, YSE has the potential to reduce methane and contribute to reduced ambient contamination. YSE is a natural feed additive that may have the potential to improve animal production.

**Author Contributions:** Conceptualization and investigation, A.Z.-S., H.B.B.-G. and J.S.-C.; Writing (original draft preparation, review and editing), A.Z.-S., R.C.A., M.E.H., H.B.B.-G., M.R.-A., Y.B.-M., N.A.S.-G., J.V.-V., F.I.-R. and J.S.-C. All authors have read and agreed to the published version of the manuscript.

**Funding:** This research received no external funding.

**Institutional Review Board Statement:** Not applicable.

**Informed Consent Statement:** Not applicable.

**Data Availability Statement:** Not applicable.

**Conflicts of Interest:** The authors declare no conflict of interest.

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
