# Peer review of "Antimicrobial and Digestive Effects of Yucca schidigera Extracts Related to Production and Environment Implications of Ruminant and Non-Ruminant Animals: A Review"

_agriculture, doi:10.3390/agriculture12081198_

Round 1

Reviewer 1 Report

General comments

I have had the opportunity to review the manuscript. The manuscript is interesting for the livestock sector. More details need to be added below my considerations.

Introduction

The structure of the introduction must be correct, it starts with point 1 and then continues with sub-points 2.1-2.2.

The introduction is well structured and described, however, it needs more detail and I suggest reading and adding the following articles:

The introduction is well structured and described, however, it needs more detail and I suggest reading and adding the following articles:

https://doi.org/10.4081/ija.2015.607

https://doi.org/10.4081/ija.2015.609

https://doi.org/10.3390/ani10030515

L 180-190 the paragraph on pigs needs to be expanded and discussed in more detail. 

L 161 and L 251 the paragraph on chickens must be discussed together.

The conclusions need to be expanded. 

Author Response

The structure of the introduction must be correct, it starts with point 1 and then continues with sub-points 2.1-2.2.

Author response: The sequence of manuscript was changed according with the observation, thanks.

The introduction is well structured and described, however, it needs more detail and I suggest reading and adding the following articles:

https://doi.org/10.4081/ija.2015.607

https://doi.org/10.4081/ija.2015.609

https://doi.org/10.3390/ani10030515

Author response: To consider these articles was necessary to include a new sentence into manuscript. Two references were added.

L 180-190 the paragraph on pigs needs to be expanded and discussed in more detail. 

Author response: we agree with the observation and the paragraph was expanded and discussed with more detail. Thanks.

L 161 and L 251 the paragraph on chickens must be discussed together.

Author response: This observation could be adequate; however, in our perception these are two different themes, and the order that they are presented into manuscript is adequate. Please note that first is presented the influence of YSE on microorganisms, and then is presented the influence on the productive behavior of animals. 

The conclusions need to be expanded. 

Author response: good observation, conclusion was expanded with more detail. Thanks

Reviewer 2 Report

The manuscript fits well within the scope of the journal. The Authors have investigated an interesting topic and the theme has been properly described. The objectives of the study were clearly defined.

I would like to underline and applaud the authors for the novelty of their research as there are not many papers dealing with these specific plant in food animal production. 

The Introduction is written concisely and provides sufficient background. The design of the review allows for making reliable conclusions.

Results of the authors on the specific thematic are well presented and thoroughly discussed and data interpretation is appropriate.

The review is well written, presented and discussed, and understandable to a specialist readership.

No significant limitations have been detected, whereas the paper presents novel and useful findings. The presented collected data have significant practical implications.

In conclusion, I recommend the acceptance of the review for publication after minor correction which is provided within the text.

All the best and stay safe,

Author Response

Thank you very much for the comments

All observations were attended, and the reference was included.

This manuscript is a resubmission of an earlier submission. The following is a list of the peer review reports and author responses from that submission.